# Oxygen systems to improve clinical care and outcomes for children and neonates: A stepped-wedge cluster-randomised trial in Nigeria

**Hamish R. Graham** [1,2]*, **Ayobami A. Bakare** [2], **Adejumoke I. Ayede** [2,3], **Amy Z. Gray** [1], **Barbara McPake** [4], **David Peel** [5], **Olatayo Olatinwo** [6], **Oladapo B. Oyewole** [2], **Eleanor F. G. Neal** [1,7], **Cattram D. Nguyen** [8,9], **Shamim A. Qazi** [10¤], **Rasa Izadnegahdar** [11], **John B. Carlin** [8], **Adegoke G. Falade** [2,3], **Trevor Duke** [1]

1 Centre for International Child Health, University of Melbourne, MCRI, Royal Children's Hospital, Parkville, Australia, 2 Department of Paediatrics, University College Hospital, Ibadan, Nigeria, 3 Department of Paediatrics, University of Ibadan, Ibadan, Nigeria, 4 Nossal Institute for Global Health, University of Melbourne, Parkville, Australia, 5 Ashdown Consultants, Hartfield, England, 6 Biomedical Services, University College Hospital, Ibadan, Nigeria, 7 Asia-Pacific Health, New Vaccines, MCRI, Royal Children's Hospital, Parkville, Australia, 8 Clinical Epidemiology and Biostatistics Unit, MCRI, Royal Children's Hospital, Parkville, Australia, 9 Department of Paediatrics, University of Melbourne, Royal Children's Hospital, Parkville, Australia, 10 Department of Maternal, Newborn, Child and Adolescent Health, World Health Organization, Geneva, Switzerland, 11 Bill and Melinda Gates Foundation, Seattle, Washington, United States of America

¤ Current address: Independent Consultant Paediatrician, Cointrin, Switzerland
* Hamish.graham@rch.org.au

**Data Availability Statement:** All dataset files are available from the OSF database: osf.io/pumdq.

## Abstract

### Background

Improving oxygen systems may improve clinical outcomes for hospitalised children with acute lower respiratory infection (ALRI). This paper reports the effects of an improved oxygen system on mortality and clinical practices in 12 general, paediatric, and maternity hospitals in southwest Nigeria.

### Methods and findings

We conducted an unblinded stepped-wedge cluster-randomised trial comparing three study periods: baseline (usual care), pulse oximetry introduction, and stepped introduction of a multifaceted oxygen system. We collected data from clinical records of all admitted neonates (<28 days old) and children (28 days to 14 years old). Primary analysis compared the full oxygen system period to the pulse oximetry period and evaluated odds of death for children, children with ALRI, neonates, and preterm neonates using mixed-effects logistic regression. Secondary analyses included the baseline period (enabling evaluation of pulse oximetry introduction) and evaluated mortality and practice outcomes on additional subgroups. Three hospitals received the oxygen system intervention at 4-month intervals. Primary analysis included 7,716 neonates and 17,143 children admitted during the 2-year stepped crossover period (November 2015 to October 2017). Compared to the pulse

**Funding:** This study was funded by the Bill and Melinda Gates Foundation (OPP1123577, awarded to TD; https://www.gatesfoundation.org/). HRG, AAB, AIA, DP, EFGN, TD, OO, OBO, and AGF received payment for services on this project from the funder (Bill and Melinda Gates Foundation), and RI is an employee of the funder. RI represented the funding agency and participated in site selection and methods meetings that informed the study design, and RI viewed the draft manuscript. The funders had no role in data collection and analysis, decision to publish, or preparation of the manuscript.

**Competing interests:** I have read the journal's policy and the authors of this manuscript have the following competing interests: HRG reports consultancy grants from WHO for unrelated work during the conduct of the study. CDN has previously received project funding from Pfizer for a vaccine impact study. All other authors declare no competing interests.

**Abbreviations:** AFE, acute febrile encephalopathy; ALRI, acute lower respiratory infection; aOR, adjusted odds ratio; CFR, case fatality rate; CI, confidence interval; CPAP, continuous positive airway pressure; CRF, case report form; ICC, intracluster correlation coefficient; IQR, interquartile range; LBW, low birth weight; LMIC, low- and middle-income country; OR, odds ratio; WHO, World Health Organization.

oximetry period, the full oxygen system had no association with death for children (adjusted odds ratio [aOR] 1.06; 95% confidence interval [CI] 0.77–1.46; $p = 0.721$) or children with ALRI (aOR 1.09; 95% CI 0.50–2.41; $p = 0.824$) and was associated with an increased risk of death for neonates overall (aOR 1.45; 95% CI 1.04–2.00; $p = 0.026$) but not preterm/low-birth-weight neonates (aOR 1.30; 95% CI 0.76–2.23; $p = 0.366$). Secondary analyses suggested that the introduction of pulse oximetry improved oxygen practices prior to implementation of the full oxygen system and was associated with lower odds of death for children with ALRI (aOR 0.33; 95% CI 0.12–0.92; $p = 0.035$) but not for children, preterm neonates, or neonates overall (aOR 0.97, 95% CI 0.60–1.58, $p = 0.913$; aOR 1.12, 95% CI 0.56–2.26, $p = 0.762$; aOR 0.90, 95% CI 0.57–1.43, $p = 0.651$). Limitations of our study are a lower-than-anticipated power to detect change in mortality outcomes (low event rates, low participant numbers, high intracluster correlation) and major contextual changes related to the 2016–2017 Nigerian economic recession that influenced care-seeking and hospital function during the study period, potentially confounding mortality outcomes.

## Conclusions

We observed no mortality benefit for children and a possible higher risk of neonatal death following the introduction of a multifaceted oxygen system compared to introducing pulse oximetry alone. Where some oxygen is available, pulse oximetry may improve oxygen usage and clinical outcomes for children with ALRI.

## Trial registration

Australian New Zealand Clinical Trials Registry: ACTRN12617000341325.

## Author summary

### Why was this study done?

- Oxygen therapy is important for many acute medical conditions, particularly among unwell children and newborns, in whom hypoxaemia (low blood oxygen) is common.
- Oxygen access and use are suboptimal in many hospitals in low- and middle-income countries.
- Improved oxygen systems may reduce deaths from pneumonia.
- To scale up oxygen in resource-limited settings, we need better information about how to improve oxygen systems and stronger evidence on the benefits of improved oxygen systems for newborns and children.

### What did the researchers do and find?

- We introduced pulse oximetry and improved oxygen systems in 12 Nigerian hospitals, aiming to provide continuous oxygen therapy for every child and neonate who needed it.

- We evaluated the impact of pulse oximetry and the improved oxygen system on care practices and clinical outcomes for >24,000 unwell newborns and children.

- We found that the improved oxygen system had no effect on outcomes for children, when compared against the introduction of pulse oximetry. However, pulse oximetry may have reduced the risk of death from pneumonia by approximately 50% compared to baseline.

- We found that the improved oxygen system was associated with increased risk of neonatal death, when compared against the introduction of pulse oximetry. However, neither pulse oximetry nor the full oxygen system had any effect on neonatal death when compared against baseline.

### What do these findings mean?

- Pulse oximetry should be central to all activities aiming to improve access to oxygen therapy.

- Where some oxygen is already available, the introduction of pulse oximetry may improve how oxygen is used and may reduce deaths from pneumonia.

- The negative results for newborns are surprising but should be interpreted cautiously. Analysis of results from individual hospitals shows significant variability in outcomes for newborns and children.

- Our study was challenged by a major economic recession, resulting in unexpected changes in admission numbers, illness severity, and care practices and low power to detect change in clinical outcomes.

## Introduction

Oxygen is an essential medical therapy that is important in resuscitation, medical, surgical, and obstetric care [1]. Oxygen therapy is particularly important for acutely unwell children and neonates, as hypoxaemia affects approximately one in five neonates and one in 10 children who are admitted to hospital [2] and increases their risk of death approximately 6-fold [3–6].

Currently, many seriously ill children in low- and middle-income countries (LMICs) lack access to oxygen therapy because of inadequate equipment, weak maintenance systems, high oxygen costs, and poor clinical practices [7–13]. Effective improvement of oxygen systems is complex, requiring context-specific technical, clinical, and managerial solutions [7].

Previous studies in low-resource settings have shown that improved oxygen systems can reduce inpatient deaths from pneumonia by up to 35% when implemented as part of a comprehensive quality-improvement program [14, 15]. If similar results were achieved from the scale-up of pulse oximetry and oxygen in the 12 countries with the largest number of child deaths, 148,000 pneumonia-related child deaths could be prevented annually [16]. Although hypoxaemia complicates many other common conditions, there are limited data on the impact of improved oxygen systems for children with non-pneumonia conditions [7, 15].

As a populous middle-income country in sub-Saharan Africa, Nigeria continues to struggle with high neonatal and child mortality rates (2017: under-5 mortality 100, neonatal mortality 33, per 1,000 live births) [17]. Recent global estimates show that Nigeria contributed one-third of under-5 malaria deaths and one-sixth of under-5 pneumonia deaths globally [18]. The conditions responsible for the largest number of child deaths in Nigeria are pneumonia (18%), malaria (14%), complications of prematurity (12%), neonatal encephalopathy and trauma (11%), diarrhoeal diseases (10%), and neonatal sepsis (5%) [18]. Data from sub-Saharan Africa suggest that hypoxaemia affects approximately 14% of children and neonates admitted to hospital, including 28%–49% of children with pneumonia, 8%–30% of children with malaria, and 22%–41% of neonates [2, 3, 6, 19–26].

The Oxygen Implementation Project was established in 2015 to evaluate an improved oxygen system in secondary health facilities in southwest Nigeria and provide evidence for future scale-up activities. The project was implemented with the support of the Gates Foundation, World Health Organization (WHO), and federal and state health agencies. Earlier findings have been published elsewhere [6, 27–29].

This paper reports the effects of an improved oxygen system on mortality and clinical practices for neonates and children under 15 years of age in 12 Nigerian hospitals.

## Methods

This study obtained ethics approval from the University of Melbourne (1543797.1) and University of Ibadan/University College Hospital Ethics Committee, Ibadan, Nigeria (UI/EC/16/0413).

### Study design

We conducted an unblinded stepped-wedge cluster-randomised trial in 12 hospitals. We introduced pulse oximetry to all hospitals in November 2015, followed by stepped introduction of the full oxygen system intervention over four sequences (three hospitals every 4 months from March 2016 to March 2017) following a prespecified timetable (Fig 1, S1 Fig). We completed postintervention data collection at the end of October 2017. We collected 2 years of preintervention data retrospectively from all hospitals for extended analyses (November 2013 to October 2015).

We chose a stepped-wedge design as the most efficient and pragmatic trial design, enabling all hospitals to receive the full oxygen system intervention over a period that was commensurate with the logistics of implementation. Clusters were at the hospital level, the natural level at which the intervention would be implemented. Geographic separation between hospitals ensured minimal risk of contamination. Additional detail on study methods has been published elsewhere [27] (protocol DOI: https://doi.org/10.1186/s13063-017-2241-8).

### Participants

We selected hospitals in collaboration with state health authorities, purposefully selecting hospitals that were representative of secondary-level health facilities and that provided inpatient care to children. Hospitals were eligible if they were registered as secondary-level health facilities and admitted at least 150 children per year. We included government and mission-run hospitals but not other private hospitals. We included hospitals in four states of southwest Nigeria (Oyo, Ondo, Ogun, and Osun states). We included nine general hospitals and three hospitals that focused on paediatric and/or maternity care (see S1 Text for additional detail on participating hospitals). AGF and AIA enrolled hospitals following an informational meeting

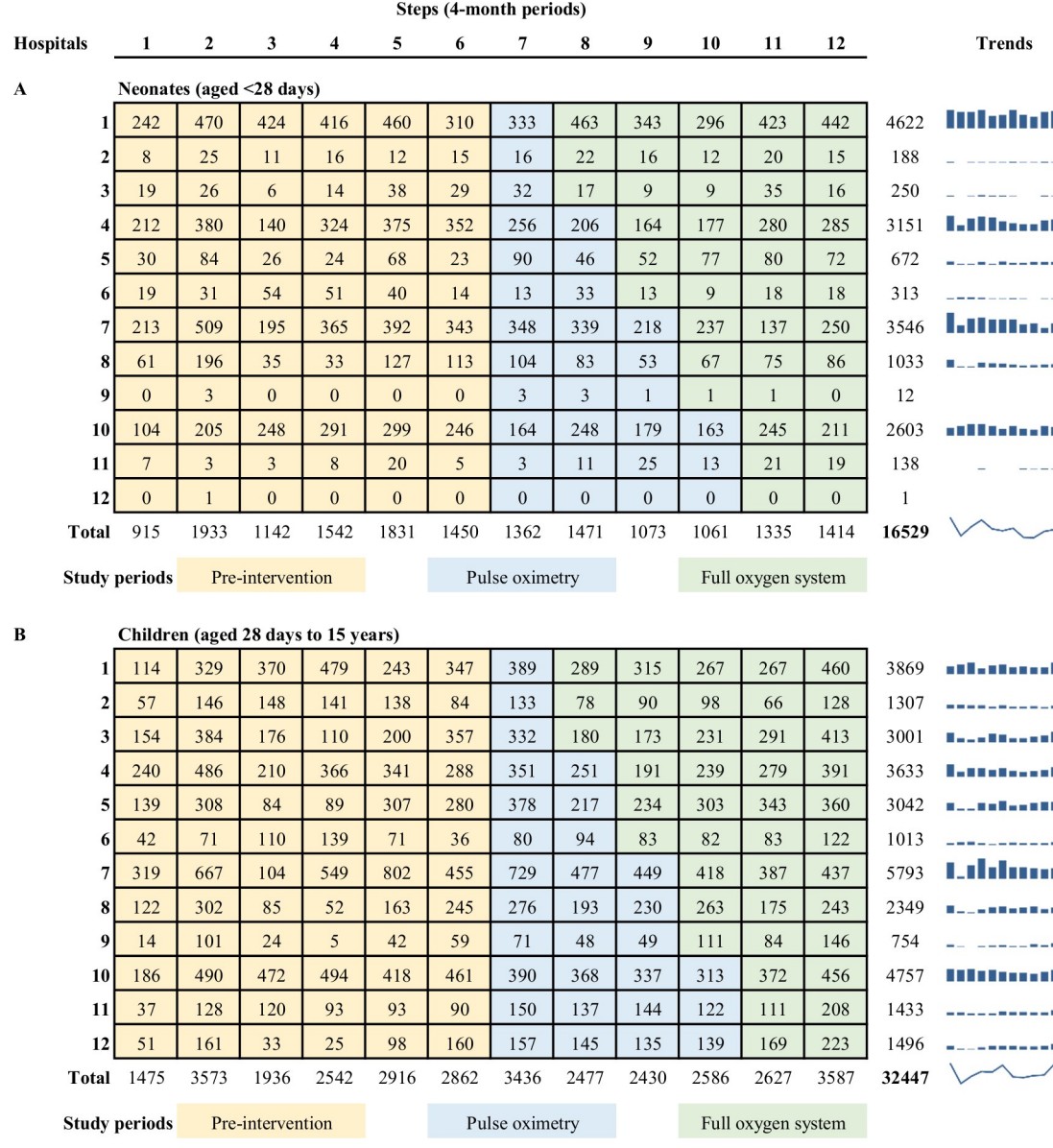

**Fig 1. Study diagram showing number of admissions in each hospital, by step.** Rows indicate individual hospitals (clusters). Columns indicate individual time periods (steps). Numbers indicate the number of neonatal (A) and child (B) participants included in analyses (excluding those admitted during the 2-week transition period at the beginning of each step). We provided all hospitals with pulse oximeters and training in November 2015 (step 7) and then introduced a multifaceted improved oxygen system to three hospitals every 4 months (randomised order). Blue bar graphs show the hospital-specific admission trends over time for quick reference, with the overall trend for neonates and children indicated with a spark-line. Notes: Data from November/December 2013 not available (first half of step 1).

with hospital representatives, with consent provided by hospital representatives prior to randomisation. We did not require individual patient/carer consent.

We collected data on all children (aged <15 years), including neonates (aged <28 days), admitted to participating hospitals during the study period. We had no additional exclusion criteria. We assumed participants admitted in different periods were independent. For effectiveness analysis, we excluded participants admitted during a 2-week 'wash-in' period at the beginning of each crossover period to implement the intervention and avoid contamination.

### Randomisation

We conducted randomisation and allocation at a team meeting in Ibadan, Nigeria, using a computer-based random sequence generator to allocate hospitals to receive the full oxygen system at prespecified dates. We stratified hospitals by size (large > 1,500 paediatric admissions per year; small/medium < 1,500) prior to randomisation, ensuring one of the four largest hospitals was represented in each step. Concealment of the intervention or its start date was not possible, as the intervention required active participation from hospital staff in planning and implementation activities.

### Intervention

The multifaceted intervention (full oxygen system) was delivered at the cluster (hospital) level and involved (1) a standardised oxygen equipment package, (2) clinical education and support, (3) technical training and support, and (4) infrastructure and systems support (Box 1, S2 Table). We designed the intervention based on experience from previous oxygen projects [7, 30, 31] and WHO guidelines [32–34] and were informed by educational, behaviour change, and implementation theories and strategies [35–39]. We adapted the intervention to local needs using a participatory planning process including a joint stakeholder meeting of representatives from all hospitals, site visits and needs assessment [13], and regular meetings

---

### Box 1. Intervention components (additional detail in S2 Table)

#### Standardised oxygen equipment package

Oxygen concentrators (Airsep Elite 5LPM), tubing and delivery devices, and maintenance materials were installed collaboratively by project and hospital technicians. (Note: Lifebox pulse oximeters were provided to all hospitals with basic training prior to the first stepped-wedge sequence.)

#### Clinical education and support

We conducted practical training based on WHO guidelines [32–34] and active learning techniques [35], led by local staff with support from the project team. Training was delivered alongside equipment installation, and hospital staff were encouraged to repeat training periodically. We provided wall charts and quick-reference summaries including simple algorithms for the clinical use of oxygen.

#### Technical training and support

We trained local hospital technicians and clinical staff to maintain equipment according to basic maintenance schedules. We provided ongoing support from project technicians to assist with troubleshooting and repairs.

#### Infrastructure and systems support

We provided a secure power supply using solar-power systems with battery storage or petrol generators. We provided technical support to hospital managers to strengthen device procurement and maintenance systems, disease surveillance and health information systems, quality-improvement processes, and oxygen financing.

---

between the project team and hospital staff. AGF, AIA, AAB, and HG conducted randomisation, invited hospital representatives to the major planning meeting, and then obtained consent to participate.

## Procedures

We commenced prospective data collection and introduced pulse oximetry to all hospitals in November 2015. We introduced pulse oximetry prior to implementation of the full oxygen system to enable quantification of hypoxaemia prevalence and evaluation of the effect of the introduction of pulse oximetry alone (extended analysis). Hospitals received the full oxygen system in March 2016, July 2016, November 2016, or March 2017, in the randomised order. We conducted retrospective data collection in late 2015, covering the preintervention period (November 2013 to October 2015) for extended analysis.

During the preintervention period, hospitals provided care as usual, including oxygen therapy using their existing supplies. During the pulse oximetry period, hospitals continued usual care with the additional availability of project-supplied pulse oximeters. Following the full oxygen system intervention, hospitals provided clinical care using the new oxygen systems, clinical guidelines, and maintenance protocols.

## Program theory

We hypothesised that to reduce mortality, effective oxygen systems must achieve (1) reliable oxygen access and (2) good clinical use of oxygen targeting hypoxaemic patients and that this would require technical, clinical, and administrative components [7]. We expected implementation of the full oxygen system to have greatest benefit for children with acute lower respiratory infection (ALRI) and preterm neonates, some effect on other conditions commonly associated with hypoxaemia (e.g., sepsis, cerebral malaria, meningitis, neonatal encephalopathy), and negligible effect on conditions rarely associated with hypoxaemia (e.g., diarrhoea, neonatal jaundice).

## Outcomes

To evaluate effectiveness, our primary outcome was death (defined as death during hospitalisation or discharged expected to die), comparing the full oxygen system period to the pulse oximetry period. We report the primary outcome for four main subgroups: (1) children (excluding neonates), (2) children with ALRI, (3) neonates, and (4) preterm and/or low birth weight (LBW) neonates.

Secondary outcomes included (1) clinical outcome of death for additional subgroups (malaria, acute febrile encephalopathy [AFE], diarrhoea, neonatal encephalopathy, and neonatal sepsis) and (2) practice outcomes relating to oxygen-related quality of care (proportion of children/neonates with documented oxygen saturation, proportion of hypoxaemic children/neonates who received oxygen therapy, proportion of children/neonates who received oxygen therapy and who had a documented indication). We analysed outcomes (1) comparing the full oxygen system period to the pulse oximetry period and (2) comparing the full oxygen system period and the pulse oximetry period to the preintervention period (see 'extended analysis' description below).

We used standard WHO case definitions to classify diagnoses (ALRI = cough or difficult breathing and any of the following: fast breathing, lower chest wall indrawing; AFE = fever and any of the following: seizures, altered conscious state; diarrhoea = >3 loose stools per day, not >14 days' duration; malaria = fever [or history of fever] and positive malaria test [rapid test or microscopy]; preterm/LBW = <37 weeks' gestational age or <2,500-g birth weight)

[34]. We based diagnostic classification on documentation in clinical records (i.e., if a particular sign was not recorded, we assumed that it was not present). Additional neonatal diagnostic classifications (neonatal sepsis, jaundice, neonatal encephalopathy) were classified according to the admission diagnosis.

All outcomes were determined using data extracted from case notes by trained data collection nurses who were based within each hospital but employed by the project. Standardised case report forms (CRFs) were completed by data collection nurses, returned to the project manager (AAB) for quality checking, and then submitted to the data manager (OBO) for double entry by trained data entry clerks using EpiData 3.1 (EpiData Association, Odense, Denmark). HG, AAB, and OBO reviewed datasets periodically in accordance with quality assurance and data management protocols [27].

## Statistical analysis

We calculated power using the Hussey and Hughes method [40, 41], basing assumptions on admission, prevalence, and mortality data from our needs assessment [13], and informed by intracluster correlation coefficient (ICC) estimates from previous studies [42]. We estimated that we would enrol 16,770 children and 7,670 neonates (1,290 preterm/LBW) during the periods of pulse oximetry and full oxygen system and based our power calculations on this time period only (without the extended preintervention period). Our original estimation contained a mathematical error, and after correction, we calculated approximately 50% power to detect a 20% reduction in overall child mortality (4% to 3.2%, ICC 0.018), 35% power to detect a 35% reduction in child pneumonia mortality (6.1% to 4.0%, ICC 0.032), 50% power to detect a 20% reduction in neonatal mortality (8.2% to 6.6%, ICC 0.063), and 55% power to detect a 25% reduction in preterm/LBW mortality (18% to 13.5%, ICC 0.123). We calculated 99% power to detect a 10-percentage-point change in proportion of hypoxaemic children who received oxygen therapy (ICC 0.05).

We described patient characteristics by study period (preintervention, pulse oximetry, and full oxygen system periods). We summarised binary variables using frequencies and percentages. We summarised continuous variables using means and standard deviations or medians and interquartile ranges (IQRs) for non-normally distributed data.

We analysed outcomes using intention-to-treat principles. We excluded participants with missing outcome data from analysis. We report intervention effects as odds ratios (OR) with 95% confidence intervals (CIs) and report the crude proportions in tabular form and visually in heat-map tables (by hospital and step).

**Analytical models.**   To evaluate effectiveness, we built mixed-effects logistic regression models fitted to individual-level binary outcome data [43, 44], following a prespecified analysis plan (S2 Text). Primary analysis compared the stepped introduction of the full oxygen system to the pulse oximetry period (steps 7–12).

Our primary analysis model included fixed effects for intervention and time (4-month steps), with random effects for cluster (hospital). To decide whether extensions to the basic model were warranted, we looked for evidence of hospital–time interaction (using a variance component for random hospital-by-time effects in an extended analysis model and comparing this with the simpler model that assumed random hospital effects only, using a likelihood ratio test) [45]. We found no evidence of hospital–time interaction for the primary clinical outcome (death). However, there was evidence of interaction for the practice outcomes, and we adjusted models accordingly.

We did 'severity-adjusted' analysis including individual signs of illness severity, age, sex, and type of hospital (mission versus government) as additional fixed effects, as we believed

clinical outcomes may be confounded by changes in admission patterns (e.g., hospitals that referred sick patients elsewhere prior to having oxygen available would keep these sicker patients, some of whom would die) (details in S2 Text).

We fitted these models to an extended time period ('extended analysis') including the pre-intervention data (steps 1–6), to evaluate the effect of pulse oximetry, and the full oxygen system against the previous standard of care. We estimated the ICC and corresponding 95% CI using Stata's 'estat icc' program after fitting the mixed-effects models. We performed all analyses using Stata 15.1 (Statacorp, College Station, TX, USA).

This trial was registered with the Australian New Zealand Clinical Trials Registry: ACTRN12617000341325 (http://www.ANZCTR.org.au/ACTRN12617000341325.aspx).

## Results

During the 2-year stepped crossover period (November 2015 and October 2017), 7,716 neonates and 17,143 children were admitted to participating hospitals and contributed to primary analysis. An additional 24,117 children (8,813 neonates and 15,304 older children) were admitted during the baseline period (from November 2013) and contributed to extended analysis. Fig 1 shows the distribution of admissions by hospital and step.

Admission numbers dropped in early 2016 in the context of economic recession, healthcare workforce industrial action, and hospital closures (Fig 1, Table 1). The distribution of participants between government and mission hospitals is unbalanced during periods of industrial action and closure of government hospitals (e.g., June/July 2016).

The distribution of patient diagnoses changed little over time, although patients tended to be slightly older during the intervention period (Table 1). Children tended to present with more severe illness during the pulse oximetry and full oxygen system periods (e.g., higher proportion with altered conscious state or WHO emergency signs).

We had complete clinical outcome data for 24,778/24,859 (99.67%) of participants in the pulse oximetry and full oxygen system periods (prospective data collection) and 23,812/24,117 (98.74%) of participants in the preintervention period (retrospective data collection). We had complete practice outcome data for 24,534/24,859 (98.69%) of participants in the pulse oximetry and full oxygen system periods and 20,677/24,117 (85.74%) of participants in the preintervention period.

### Fidelity and variation from protocol

The intervention was deployed as planned, with equipment installation and training occurring over 2–4 days at each hospital, usually in the first 2 weeks of the designated installation month (S2 Table). Solar-power installations were delayed because of technical problems with the initial solar installations (in hospitals H1, H2, and H3) requiring rectification, system redesign, and change of provider. This delayed subsequent solar installations and required temporary use of alternative power sources (mains power and/or generators). Despite this setback, all intervention hospitals had continuous access to oxygen therapy from the newly installed systems throughout the intervention period.

The project team visited each hospital at least once every 3 months and had regular communication by phone and email. Hospital staff generally reported equipment problems promptly, and our project team provided prompt assistance (including repair or replacement), ensuring patients had continuous access to oxygen therapy.

Hospitals took up some, but not all, aspects of the project's support for systems. Most hospitals started multidisciplinary 'oxygen team' meetings. Few hospitals improved their medical record keeping. All hospitals continued to charge patients for oxygen, and most did not adopt

**Table 1. Population characteristics, by study period.**

| Population characteristic | Preintervention | Pulse Oximetry | Full O₂ system |
|---|---|---|---|
| | ($n$ = 24,117) | ($n$ = 10,267) | ($n$ = 14,592) |
| Neonate (≤28 days old) | 8,813 (36.5) | 2,983 (29.1) | 4,733 (32.4) |
| Infant (1–12 months old) | 4,709 (19.5) | 2,218 (21.6) | 2,520 (17.3) |
| Young child (1–5 years old) | 7,788 (32.3) | 3,580 (34.9) | 4,803 (32.9) |
| Older child (5–15 years old) | 2,559 (10.6) | 1,452 (14.1) | 2,458 (16.8) |
| Child (age unknown) | 248 (1.0) | 34 (0.3) | 78 (0.5) |
| Age, months median (IQR) | 9.0 (0.1–24.0) | 11.1 (0.3–31.0) | 12.0 (0.1–36.0) |
| Age, months mean (SD) | 21.0 (32.2) | 24.8 (35.7) | 27.0 (37.2) |
| Sex, % female | 43.9 | 43.2 | 44.2 |
| Hospital type, % government | 57.7 | 62.4 | 57.6 |
| Hospital size, % small | 11.9 | 16.8 | 14.2 |
| **Child diagnoses and presenting signs[α]** | | | |
| ALRI | 1,897 (12.4) | 883 (12.1) | 1,269 (12.9) |
| Malaria | 5,230 (34.3) | 2,609 (35.9) | 3,835 (38.9) |
| AFE | 2,822 (18.5) | 1,578 (21.7) | 2043 (20.7) |
| Diarrhoea | 1,732 (11.3) | 1,053 (14.5) | 1,013 (10.3) |
| Malnutrition | 237 (1.7) | 184 (2.6) | 192 (2.0) |
| HIV | 18 (0.1) | 10 (0.1) | 13 (0.1) |
| Severe respiratory distress | 1,072 (8.4) | 610 (8.7) | 742 (7.5) |
| Central cyanosis | 32 (0.3) | 26 (0.4) | 70 (0.7) |
| Coma or barely conscious | 685 (5.4) | 338 (4.8) | 728 (7.4) |
| Severe dehydration | 488 (5.3) | 301 (5.4) | 320 (4.4) |
| Unable to feed | 2,166 (17.1) | 1,315 (18.6) | 1956 (19.9) |
| Any WHO emergency sign | 4,505 (35.4) | 2,747 (38.9) | 3,864 (39.3) |
| Hypoxaemia (SpO₂ < 90%) | 46/500 (9.2)[§] | 424/4,274 (9.9) | 928/9,057 (10.2) |
| **Neonatal diagnoses and presenting signs[α]** | | | |
| Preterm/LBW | 1,883 (26.8) | 688 (26.4) | 1,137 (26.7) |
| Neonatal sepsis | 3,916 (48.1) | 1,705 (59.4) | 2,297 (51.9) |
| Neonatal encephalopathy | 3,439 (42.3) | 1,071 (37.3) | 1,882 (42.6) |
| Jaundice | 2001 (24.6) | 799 (27.8) | 938 (21.2) |
| Severe respiratory distress | 617 (7.8) | 359 (12.4) | 548 (11.6) |
| Central cyanosis | 230 (2.9) | 101 (3.5) | 103 (2.2) |
| Coma or barely conscious | 230 (2.9) | 120 (4.1) | 189 (4.0) |
| Unable to feed | 1,417 (17.9) | 485 (16.7) | 927 (19.6) |
| Hypoxaemia (SpO₂ < 90%) | 30/189 (15.9)[§] | 425/1,474 (28.8) | 993/3,479 (28.5) |

Data are $n$ (%) unless otherwise indicated. Diarrhoea is defined as >3 stools per day, not >14 days' duration. Malaria is defined as fever (or history of fever) and positive malaria test (rapid test or microscopy). Preterm is defined as <37 weeks' gestational age. Other diagnoses as per admission diagnosis.

[α]Multiple diagnoses permitted.

[§]Limited hypoxaemia data available in preintervention period, mostly from a single hospital.

Abbreviations: AFE, acute febrile encephalopathy, defined as fever and one of the following: seizures, altered conscious state; ALRI, acute lower respiratory infection, defined as cough or difficult breathing and any of the following: fast breathing, lower chest wall indrawing; CFR, case fatality rate; CI, confidence interval; IQR, interquartile range; LBW, low birth weight, <2,500 g

measures to reduce costs to patients until late 2017. During the study period, Nigeria experienced an economic recession, resulting in reduced income for hospitals, unpaid healthcare worker wages, and associated industrial action [46]. Although hospital staff and administrators insisted that these financial pressures did not impede patient access to oxygen, we received

**Table 2. Effect of the full oxygen system intervention on mortality—Primary outcome analysis comparing the full oxygen system intervention to the prior standard of care plus pulse oximetry.**

| Population and study period | Deaths | Mixed-Model Adjusted Odds Ratio (95% CI)[α] | | | | ICC |
|---|---|---|---|---|---|---|
| | n/N (%)* | Basic Model | | Severity-Adjusted | | (95% CI) |
| | | | p-Value | | p-Value | |
| Child[β] | | | | | | |
| - Pulse oximetry | 227/6440 (3.5) | - | | - | | 0.03 |
| - Full O₂ system | 422/9203 (4.6) | 1.06 (0.77–1.46) | 0.721 | 1.12 (0.79–1.59) | 0.527 | (0.01–0.07) |
| Child ALRI[β] | | | | | | |
| - Pulse oximetry | 44/759 (5.8) | - | | - | | 0.16 |
| - Full O₂ system | 97/1182 (8.2) | 1.09 (0.50–2.41) | 0.824 | 1.42 (0.60–3.36) | 0.427 | (0.06–0.36) |
| Neonate | | | | | | |
| - Pulse oximetry | 204/2627 (7.8) | - | | - | | 0.11 |
| - Full O₂ system | 486/4360 (11.1) | 1.45 (1.04–2.00) | 0.026 | 1.76 (1.22–2.54) | 0.002 | (0.04–0.28) |
| Preterm/LBW neonate | | | | | | |
| - Pulse oximetry | 82/611 (13.4) | - | | - | | 0.07 |
| - Full O₂ system | 203/1042 (19.5) | 1.30 (0.76–2.23) | 0.366 | 1.74 (0.97–3.11) | 0.062 | (0.02–0.24) |

Preterm defined as <37 weeks' gestational age.

*Denominators vary according to the population included.

[α]Mixed-model odds ratios account for the clustering of patients within hospitals and adjust for time trends. Under the stepped-wedge design, the adjusted odds ratios are calculated with the use of all data points in the intervention period versus the comparison period and therefore represent the average odds of exposure to the intervention.

[β]Child under 15 years of age, excluding neonates (<28 days of age).

Abbreviations: ALRI, acute lower respiratory infection, defined as cough or difficulty breathing and any of the following: fast breathing, lower chest wall indrawing; CI, confidence interval; ICC, intracluster correlation coefficient, indicating the degree of outcome similarity within clusters (hospitals); LBW, low birth weight, <2,500 g

anecdotal reports of more patients becoming indebted to the hospital and leaving against medical advice.

## Intervention effects

**Primary clinical outcomes.** Table 2 shows the crude rates and intervention effect sizes for the primary analysis of clinical outcomes.

Primary analysis suggests that, compared to the pulse oximetry period, the full oxygen system was not associated with death for children overall (aOR 1.06; 95% CI 0.77–1.46; $p$ = 0.721) or children with ALRI (aOR 1.09; 95% CI 0.50–2.41; $p$ = 0.824) and was associated with increased risk of death for neonates overall (aOR 1.45; 95% CI 1.04–2.00; $p$ = 0.026) but not preterm/LBW neonates (aOR 1.30; 95% CI 0.76–2.23; $p$ = 0.366). Severity-adjusted analysis was qualitatively similar to the basic model, with stronger evidence of harm for neonates (Table 2).

At reviewer request, we conducted post hoc analysis on subgroups with hypoxaemia and children under 5 years of age, with qualitatively similar results (S3 Table).

**Secondary clinical outcomes and extended analysis of preintervention period.** Table 3 shows the crude rates and intervention effect sizes for the primary and secondary clinical outcomes, including the extended analysis comparing pulse oximetry and full oxygen periods against the preintervention period.

Compared to the preintervention period, the pulse oximetry and full oxygen system intervention were associated with decreased risk of death for children with ALRI (aOR 0.45, 95% CI 0.17–1.18, $p$ = 0.104; aOR 0.57, 95% CI 0.31–1.04, $p$ = 0.066), with qualitatively similar

**Table 3. Effect of the intervention(s) on mortality, showing the primary analysis (full oxygen period versus pulse oximetry period) and extended analysis (comparing pulse oximetry and full oxygen system periods to the preintervention period).**

| Population and study period | Deaths n/N (%)* | Mixed-Model Adjusted Odds Ratio (95% CI) α | | | | | | | | ICCβ (95% CI) |
|---|---|---|---|---|---|---|---|---|---|---|
| | | Primary Analysis‡ | | | | Extended Analysis§ | | | | |
| | | Basic Model | p | Severity-Adjusted | p | Basic Model | p | Severity-Adjusted | p | |
| **Child** | | | | | | | | | | |
| - Preintervention | 632/15,067 (4.2) | | | | | - | | - | | |
| - Pulse oximetry | 227/6,440 (3.5) | - | | - | | 1.10 (0.72–1.69) | 0.659 | 0.97 (0.60–1.58) | 0.913 | 0.05 |
| - Full O₂ system | 422/9,203 (4.6) | 1.06 (0.77–1.46) | 0.721 | 1.12 (0.79–1.59) | 0.527 | 1.05 (0.77–1.42) | 0.779 | 1.03 (0.72–1.47) | 0.861 | (0.02–0.10) |
| **Child ALRI** | | | | | | | | | | |
| - Preintervention | 116/1,887 (6.1) | | | | | - | | - | | |
| - Pulse oximetry | 44/759 (5.8) | - | | - | | 0.45 (0.17–1.18) | 0.104 | 0.33 (0.12–0.92) | 0.035 | 0.10 |
| - Full O₂ system | 97/1,182 (8.2) | 1.09 (0.50–2.41) | 0.824 | 1.42 (0.60–3.36) | 0.427 | 0.57 (0.31–1.04) | 0.066 | 0.50 (0.26–0.98) | 0.044 | (0.04–0.22) |
| **Neonate** | | | | | | | | | | |
| - Preintervention | 913/8,745 (10.4) | | | | | - | | - | | |
| - Pulse oximetry | 204/2,627 (7.8) | - | | - | | 0.89 (0.60–1.32) | 0.561 | 0.90 (0.57–1.43) | 0.651 | 0.13 |
| - Full O₂ system | 486/4,360 (11.1) | 1.45 (1.04–2.00) | 0.026 | 1.76 (1.22–2.54) | 0.002 | 0.97 (0.74–1.27) | 0.811 | 0.90 (0.65–1.24) | 0.514 | (0.05–0.32) |
| **Preterm/LBW neonate** | | | | | | | | | | |
| - Preintervention | 326/1,876 (17.4) | | | | | - | | - | | |
| - Pulse oximetry | 82/611 (13.4) | - | | - | | 1.41 (0.74–2.68) | 0.293 | 1.12 (0.56–2.26) | 0.762 | 0.10 |
| - Full O₂ system | 203/1,042 (19.5) | 1.30 (0.76–2.23) | 0.366 | 1.74 (0.97–3.11) | 0.062 | 1.05 (0.69–1.60) | 0.829 | 0.99 (0.61–1.59) | 0.957 | (0.04–0.25) |
| **Additional subgroups** | | | | | | | | | | |
| **Child AFE** | | | | | | | | | | |
| - Preintervention | 203/2,810 (7.2) | | | | | - | | - | | |
| - Pulse oximetry | 97/1,418 (6.8) | - | | - | | 1.10 (0.54–2.23) | 0.800 | 1.01 (0.47–2.17) | 0.982 | 0.09 |
| - Full O₂ system | 192/1,885 (10.2) | 1.06 (0.65–1.74) | 0.803 | 1.04 (0.61–1.77) | 0.895 | 1.05 (0.62–1.79) | 0.853 | 0.92 (0.52–1.64) | 0.785 | (0.04–0.19) |
| **Child malaria** | | | | | | | | | | |
| - Preintervention | 133/5,201 (2.6) | | | | | - | | - | | |
| - Pulse oximetry | 52/2,298 (2.3) | - | | - | | 0.79 (0.33–1.93) | 0.611 | 0.79 (0.31–1.99) | 0.616 | 0.05 |
| - Full O₂ system | 114/3,593 (3.2) | 1.61 (0.88–2.97) | 0.124 | 1.42 (0.75–2.71) | 0.281 | 1.15 (0.59–2.24) | 0.686 | 1.10 (0.55–2.21) | 0.790 | (0.02–0.14) |
| **Child diarrhoea** | | | | | | | | | | |
| - Preintervention | 56/1,726 (3.2) | | | | | - | | - | | |
| - Pulse oximetry | 27/954 (2.8) | - | | - | | 1.70 (0.44–6.60) | 0.440 | 2.77 (0.60–12.87) | 0.194 | 0.16 |
| - Full O₂ system | 43/944 (4.6) | 0.98 (0.36–2.68) | 0.964 | 0.72 (0.22–2.34) | 0.579 | 1.58 (0.62–4.01) | 0.335 | 1.78 (0.62–5.14) | 0.284 | (0.06–0.40) |
| **Neonatal sepsis** | | | | | | | | | | |
| - Preintervention | 369/3,893 (9.5) | | | | | - | | - | | |
| - Pulse oximetry | 108/1,494 (7.2) | - | | - | | 1.00 (0.56–1.77) | 0.992 | 0.86 (0.45–1.64) | 0.656 | 0.13 |
| - Full O₂ system | 254/2,128 (11.9) | 1.65 (1.06–2.59) | 0.028 | 1.90 (1.16–3.11) | 0.010 | 1.09 (0.73–1.65) | 0.667 | 0.90 (0.57–1.43) | 0.660 | (0.04–0.32) |
| **Neonatal encephalopathy** | | | | | | | | | | |
| - Preintervention | 397/3,423 (11.6) | | | | | - | | - | | |
| - Pulse oximetry | 107/943 (11.3) | - | | - | | 0.95 (0.53–1.71) | 0.863 | 0.88 (0.45–1.72) | 0.703 | 0.13 |
| - Full O₂ system | 242/1,736 (13.9) | 1.46 (0.92–2.34) | 0.110 | 2.01 (1.27–3.20) | 0.003 | 0.98 (0.65–1.47) | 0.911 | 0.90 (0.56–1.45) | 0.667 | (0.05–0.31) |
| **Neonatal jaundice** | | | | | | | | | | |
| - Preintervention | 131/1,991 (6.6) | | | | | - | | - | | |
| - Pulse oximetry | 18/704 (2.6) | - | | - | | 0.91 (0.17–4.79) | 0.911 | 0.96 (0.16–5.70) | 0.965 | 0.12 |
| - Full O₂ system | 49/873 (5.6) | 3.75 (1.28–11.02) | 0.016 | 4.22 (1.07–16.59) | 0.039 | 1.89 (0.49–7.28) | 0.354 | 1.57 (0.37–6.67) | 0.544 | (0.04–0.30) |

Diarrhoea defined as >3 stools per day, not >14 days duration. Malaria defined as fever (or history of fever) and positive malaria test (rapid test or microscopy).

Neonatal sepsis, neonatal encephalopathy, and neonatal jaundice classified as per admission diagnosis. Preterm defined as <37 weeks' gestational age.

*Denominators vary according to the population included.

‡Primary analysis compares full oxygen system and pulse oximetry periods.

§Extended analysis compares pulse oximetry and full oxygen system periods to the preintervention period.

αMixed-model odds ratios account for the clustering of patients within hospitals and adjust for time trends. Under the stepped-wedge design, the adjusted odds ratios are calculated with the use of all data points in the intervention period versus the comparison period and therefore represent the average odds of exposure to the intervention.

βICC including all periods.

γChild under 15 years of age, excluding neonates (<28 days of age).

Abbreviations: AFE, acute febrile encephalopathy, defined as fever and one of the following: seizures, altered conscious state; ALRI, cough or difficulty breathing and any of the following: fast breathing, lower chest wall indrawing; CI, confidence interval; ICC, intracluster correlation coefficient; LBW, low birth weight, <2,500 g

results from severity-adjusted analysis (aOR 0.33, 95% CI 0.12–0.92, $p$ = 0.035; aOR 0.50, 95% CI 0.26–0.98, $p$ = 0.044). Neither pulse oximetry nor the full oxygen system was associated with death for children overall, neonates overall, or any of the other subgroups, compared to the preexisting standard of care.

**Practice outcomes.** Table 4 shows the crude rates and intervention effect sizes for the practice outcomes, including the extended analysis across all three study periods. Use of pulse oximetry in the clinical care of hospitalised children improved dramatically across the study periods (from <5% of admissions to >90%), with lesser change evident for oxygen practices.

For neonates, there was a statistically nonsignificant increase in overall oxygen use (aOR 1.34; 95% CI 0.96–1.87; $p$ = 0.085) following introduction of the full oxygen system, with no change in oxygen administration for neonates with hypoxaemia (aOR 0.77; 95% CI 0.34–1.75; $p$ = 0.531) compared to the pulse oximetry period. Pulse oximetry had minimal impact on overall oxygen use (aOR 1.29; 95% CI 0.82–2.03; $p$ = 0.264) or the use of oxygen for neonates with signs of hypoxaemia (aOR 1.06; 95% CI 0.54–20.9; $p$ = 0.857) compared to the preintervention period.

For children outside the neonatal period, results suggest that the full oxygen system increased overall oxygen use (aOR 1.89; 95% CI 1.34–2.66; $p$ < 0.001) and improved oxygen administration for those with hypoxaemia (aOR 2.71; 95% CI 1.12–6.56; $p$ = 0.027) compared to the pulse oximetry period. Pulse oximetry had minimal impact on overall oxygen use (aOR 0.72; 95% CI 0.43–1.19; $p$ = 0.200) or the use of oxygen for those with signs of hypoxaemia (aOR 1.00; 95% CI 0.51–1.95; $p$ = 0.995).

Heat-map graphs suggest that pulse oximetry and oxygen practice changes were achieved incrementally over a number of months, with most of this change occurring early in the pulse oximetry period (Fig 2). Of note, the proportion of hypoxaemic ($SpO_2$ < 90%) children receiving oxygen therapy increased to >75% coverage within 4–8 months of receiving pulse oximeters. Three hospitals are outliers (H9, H12, and H5's children's ward), being the only hospitals that genuinely had no oxygen access for children prior to the intervention and showed no improvement in oxygen use until after the full oxygen system was implemented.

Heat-map graphs suggest oxygen use was better targeted at neonates and children with hypoxaemia in the pulse oximetry period, but this is more weakly reflected in the adjusted effect estimates (aOR 2.00, 95% CI 0.80–4.95, $p$ = 0.136 and aOR 1.21, 95% CI 0.55–2.68, $p$ = 0.634 for neonates and children, respectively) (Fig 2, Table 4). The 'indication for oxygen' outcome uses a composite measure of hypoxaemia ($SpO_2$ < 90% on admission, or signs of hypoxaemia if no $SpO_2$ documented), which may be biased because of the dramatic change in $SpO_2$ documentation over the study periods (i.e., reflecting patients with signs of hypoxaemia in the preintervention period and patients with $SpO_2$ < 90% in the latter periods).

## Discussion

This study reports results from the first cluster-randomised trial of improved oxygen systems. We found that the improved oxygen system had no effect on outcomes for children when compared against the introduction of pulse oximetry. However, pulse oximetry may have reduced the risk of death from pneumonia by approximately 50% compared to baseline. We found that the improved oxygen system was associated with increased risk of neonatal death when compared against the introduction of pulse oximetry. However, neither pulse oximetry nor the full oxygen system had any effect on neonatal death when compared against baseline.

As with all stepped-wedge trials, this study is complex in design and analysis, and interpretation of the estimated treatment effects must be made in the context of the study and with

**Table 4. Effect of the intervention(s) on practice outcomes, showing the primary analysis (full oxygen period versus pulse oximetry period) and extended analysis (comparing pulse oximetry and full oxygen system periods to the preintervention period).**

| Population and study period | Proportions | Mixed-Model Adjusted Odds Ratio (95% CI)[α] | | | | ICC[β] |
|---|---|---|---|---|---|---|
| | n/N (%)* | Primary Analysis[‡] | | Extended Analysis[§] | | (95% CI) |
| | | | p-Value | | p-Value | |
| **Neonates** | | | | | | |
| Pulse oximetry | | | | | | |
| - Preintervention | 219/7,940 (2.8%) | | | - | | |
| - Pulse oximetry | 1,723/2,569 (67.1%) | - | | 15,115 (1,383–165,254) | <0.001 | 0.67 |
| - Full O₂ system | 4,122/4,367 (94.4%) | 3.31 (0.99–11.12) | 0.053 | 63,826 (8,318–189,743) | <0.001 | (0.46–0.83) |
| Oxygen (to anyone) | | | | | | |
| - Preintervention | 1,794/7,940 (22.6%) | | | - | | |
| - Pulse oximetry | 559/2,569 (21.8%) | - | | 1.29 (0.82–2.03) | 0.264 | 0.07 |
| - Full O₂ system | 1,416/4,367 (32.4%) | 1.34 (0.96–1.87) | 0.085 | 2.03 (1.46–2.82) | <0.001 | (0.03–0.16) |
| Oxygen if SpO₂ < 90% | | | | | | |
| - Preintervention | 24/30 (80.0%) | | | - | | |
| - Pulse oximetry | 307/380 (80.8%) | - | | 1.86 (0.18–18.88) | 0.598 | 0.01 |
| - Full O₂ system | 833/921 (90.4%) | 0.77 (0.34–1.75) | 0.531 | 1.46 (0.16–13.04) | 0.734 | (0.00–0.22) |
| Oxygen if signs of hypoxaemia | | | | | | |
| - Preintervention | 1,036/1,955 (53.0%) | | | - | | |
| - Pulse oximetry | 345/658 (52.4%) | - | | 1.06 (0.54–20.9) | 0.857 | 0.02 |
| - Full O₂ system | 693/1,132 (61.2%) | 1.14 (0.78–1.67) | 0.502 | 1.43 (0.87–2.36) | 0.160 | (0.01–0.06) |
| Indication for oxygen[¥] | | | | | | |
| - Preintervention | 1,002/1,794 (55.9%) | | | - | | |
| - Pulse oximetry | 365/559 (65.3%) | - | | 2.00 (0.80–4.95) | 0.136 | 0.07 |
| - Full O₂ system | 858/1,416 (60.6%) | 1.42 (0.77–2.63) | 0.265 | 2.09 (1.09–4.02) | 0.027 | (0.02–0.19) |
| **Children** | | | | | | |
| Pulse oximetry | | | | | | |
| - Preintervention | 500/12,737 (3.9%) | | | - | | |
| - Pulse oximetry | 3,941/6,298 (62.6%) | - | | 1,584 (179–13,994) | <0.001 | 0.28 |
| - Full O₂ system | 8,463/9,216 (91.8%) | 6.60 (2.52–17.29) | <0.001 | 9,454 (2,309–38,715) | <0.001 | (0.15–0.46) |
| Oxygen (to anyone) | | | | | | |
| - Preintervention | 1,184/12,737 (9.3%) | | | - | | |
| - Pulse oximetry | 455/6,298 (7.2%) | - | | 0.72 (0.43–1.19) | 0.200 | 0.14 |
| - Full O₂ system | 1,262/9,216 (13.7%) | 1.89 (1.34–2.66) | <0.001 | 1.50 (1.04–2.15) | 0.028 | (0.06–0.28) |
| Oxygen if SpO2<90% | | | | | | |
| - Preintervention | 34/46 (73.9%) | | | - | | |
| - Pulse oximetry | 265/395 (67.1%) | - | | 2.85 (0.19–43.63) | 0.451 | 0.12 |
| - Full O₂ system | 694/851 (81.6%) | 2.71 (1.12–6.56) | 0.027 | 6.90 (0.52–91.70) | 0.143 | (0.04–0.33) |
| Oxygen if signs of hypoxaemia | | | | | | |
| - Preintervention | 810/4,244 (19.1%) | | | - | | |
| - Pulse oximetry | 327/1,890 (17.3%) | - | | 1.00 (0.51–1.95) | 0.995 | 0.11 |
| - Full O₂ system | 755/2,265 (33.3%) | 1.63 (1.05–2.53) | 0.030 | 1.71 (1.06–2.75) | 0.028 | (0.04–0.24) |
| Indication for oxygen[¥] | | | | | | |
| - Preintervention | 778/1,184 (65.7%) | | | - | | |
| - Pulse oximetry | 311/455 (68.4%) | - | | 1.21 (0.55–2.68) | 0.634 | 0.03 |
| - Full O₂ system | 727/1,262 (57.6%) | 0.83 (0.51–1.35) | 0.457 | 0.73 (0.41–1.31) | 0.291 | (0.01–0.11) |

Data are n/N (%) unless otherwise indicated.

*Denominators vary according to the population included.

[‡]Primary analysis compares full oxygen system and pulse oximetry periods.

[§]Extended analysis compares pulse oximetry and full oxygen system periods to the preintervention period.

[α]Mixed-model odds ratios account for the clustering of patients within hospitals and adjust for time trends. Under the stepped-wedge design, the adjusted odds ratios are calculated with the use of all data points in the intervention period versus the comparison period and therefore represent the average odds of exposure to the intervention.

[β]ICC including all periods.

[¥]'Indication for oxygen' is calculated as [# with SpO₂ < 90% on admission (or signs of hypoxaemia if no SpO₂ documented)/# prescribed oxygen therapy]. Given the dramatic change in SpO₂ documentation over the study periods, this outcome measure may be biased.

Abbreviations: CI, confidence interval; ICC, intracluster correlation coefficient

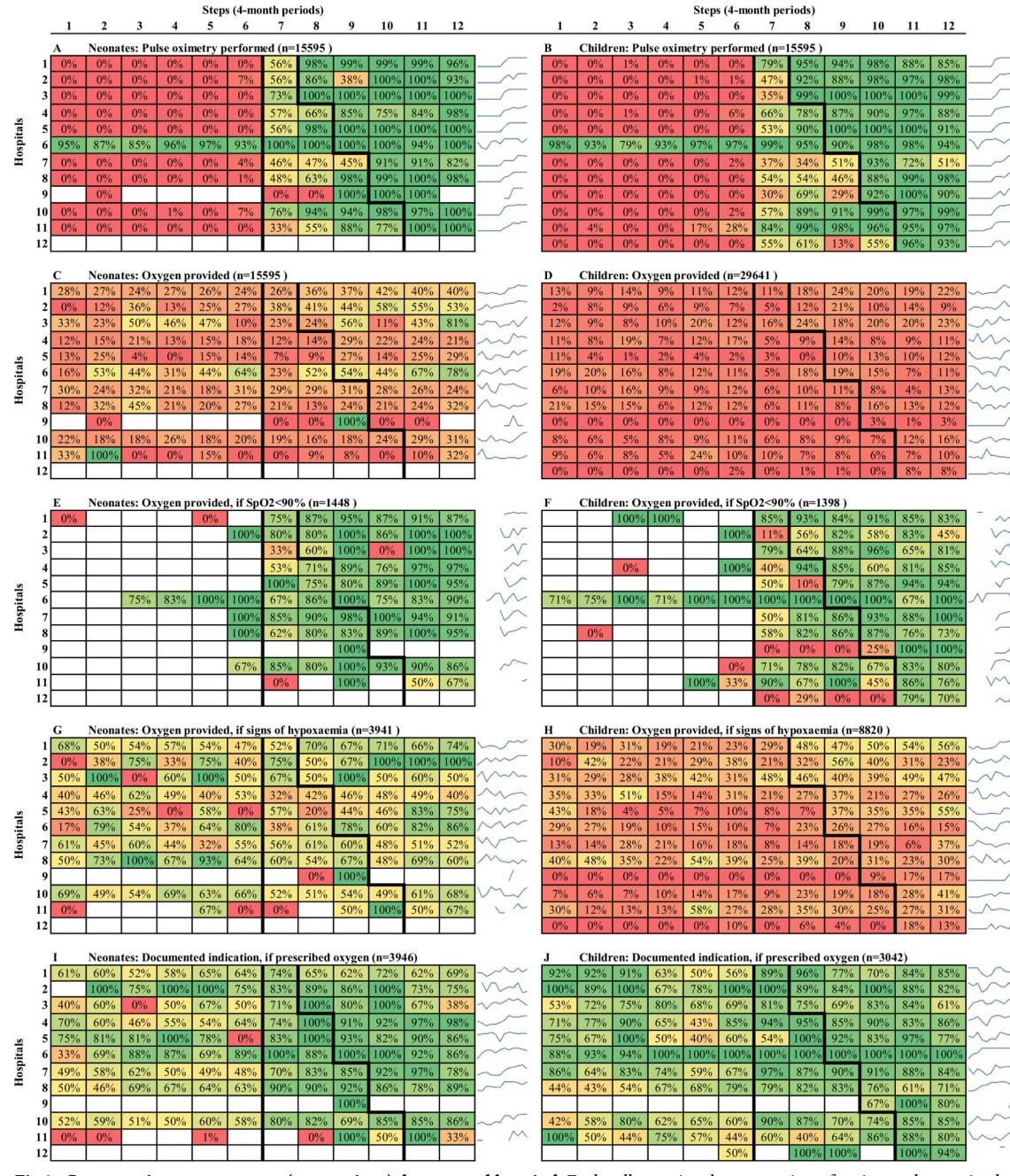

**Fig 2. Care practice outcome rates (proportions), by step and hospital.** Each cell contains the proportion of patients who received a particular element of care (pulse oximetry or oxygen therapy) on admission. For each of the 10 panels in this figure, the 12 hospitals are represented on the y-axis, whereas the steps of the trial are represented on the x-axis. The colour gradient extends from red (0%) through yellow to green (100%), providing a visual representation of practice change over time. Blue spark-lines show the hospital-specific trend for easy reference. When the denominator to compute the cell rate is 0, cells are coloured in white. Notes: Data from November/December 2013 not available in step 1.

consideration to potential additional confounders. With that in mind, we offer two brief comments before discussing the results.

Firstly, this study was not an individual trial of oxygen versus no oxygen. Rather, the introduction of an improved oxygen system was compared to a previous standard of care

augmented with pulse oximetry. At baseline, this previous standard of care included a functional oxygen cylinder or concentrator in 11/12 hospitals and in the paediatric area of 6/12 hospitals [29].

Secondly, during the study period, Nigeria experienced a major economic recession, which impacted individuals, households, health facilities, and the broader health system [46]. Participating government hospitals had extended periods of unpaid salaries, resulting in industrial action, hospital closure, increases in user fees, and deterioration in morale [28]. In a user-pay system, economic hardship for staff and patients likely contributed to the lower admission numbers and increased severity of presenting illness during the intervention period and possibly impacted quality of care and clinical outcomes.

The far-reaching impact of the economic recession may have introduced confounding that could not be accounted for in analyses. Our primary analysis model for effectiveness included adjustment for clustering (accounting for hospital-specific influences on outcomes) and time (accounting for step-specific influences on outcomes). Our severity-adjusted analysis model adjusted for additional patient-specific factors, including age, signs of severe illness, and diagnosis. In the context of the recession, the severity-adjusted model may represent a better estimate of intervention effect.

## Interpreting results

We will now explore the clinical and practice outcomes in light of our program theory and the broader study context to provide a balanced and informed interpretation of results.

**Clinical impact for children.** We expected the improved oxygen system to have the most benefit for children with ALRI. We found no mortality benefit for children with ALRI or for other subgroups of interest in primary analysis (comparing the full oxygen system to the pulse oximetry period), although the study was underpowered to do so. However, extended analysis suggested that the improved oxygen system realised a mortality benefit for children with ALRI compared to the baseline period and that it was fully realised through the introduction of pulse oximetry.

These findings are consistent with our practice outcome results. Our baseline assessment suggested that oxygen supplies were limited in all participating hospitals, with only half of hospitals having oxygen supplies available for children or neonates at the time of assessment [13]. However, the larger hospitals typically had a better supply and, since they contributed more observations to our current study overall results, may be more reflective of this scenario. Furthermore, practice data suggest that most hospitals significantly improved their oxygen practices during the pulse oximetry period, reaching high levels of oxygen therapy administration for hypoxaemic children prior to the full oxygen system period.

This leads us to two likely interpretations. Firstly, when oxygen is available, pulse oximetry enables healthcare workers to better target it to those who are hypoxaemic. Indeed, even though oxygen supplies are insufficient to meet the needs of all hypoxaemic children, pulse oximetry may improve oxygen practices sufficiently to reach those who would benefit the most (i.e., with more severe hypoxaemia) with less opportunity for further mortality reduction with enhanced oxygen supplies. Secondly, when oxygen supplies are limited, objective evidence of high hypoxaemia prevalence, through the use of pulse oximetry, enables hospitals to mobilise additional oxygen supplies and prioritise paediatric patients. Anecdotal feedback from hospitals staff supports this theory, with only two small hospitals (H9 and H12) unable to mobilise additional oxygen from other wards and remaining without oxygen for the entire pulse oximetry period.

These theories are supported by previous before–after studies in two Papua New Guinean hospitals that reported ALRI mortality reductions with the introduction of pulse oximetry

(4.2% down to 2.7% and risk ratio 0.65; 95% CI 0.41–1.02) [12, 47] in the context of reasonable access to oxygen cylinders at baseline (e.g., oxygen available for 87% of children with $SpO_2 <$ 90% [12]). These effect sizes are similar to our findings (aOR 0.57; 95% CI 0.31–1.04; $p$ = 0.066) and to previous before–after studies involving the introduction of pulse oximetry and improved oxygen systems [14, 15, 48]. Of these, the highest-quality data come from the five-hospital study from Papua New Guinea, which found that an improved oxygen system reduced inpatient child pneumonia case fatality rates (CFRs) by 35% (risk ratio 0.65; 95% CI 0.52–0.78) [14].

We did not find evidence of a mortality benefit for children overall, in either the primary analysis or the extended analysis. Previous studies have reported reductions in non-pneumonia CFRs following introduction of an improved oxygen system, including the study in Papua New Guinea, which found non-pneumonia CFRs were reduced by 21% (risk ratio 0.79; 95% CI 0.70–0.89) [31, 49].

**Clinical impact for neonates.**　We expected the improved oxygen system to have the most benefit for preterm neonates. Surprisingly, our primary analysis suggested that the full oxygen system may have increased the odds of death for neonates overall (aOR 1.45; 95% CI 1.04–2.00), although there was no significant increase in the preterm/LBW neonate subgroup (aOR 1.30; 95% CI 0.76–2.23). In contrast, extended analysis suggested no effect of either pulse oximetry or the full oxygen system on odds of neonatal death when compared to the baseline period. The latter would fit with practice outcome findings of high oxygen use for neonates in the larger hospitals at baseline (and little room for improvement).

It is possible that the finding of harm to neonates may be due to chance, given the multiple analyses and discordance in results between the primary analysis and the extended and secondary analyses. However, there are other potential explanations that warrant consideration.

First, we know that excessive oxygen administration can cause harm. Oxygen therapy can increase the production of reactive oxygen species, which can overwhelm the body's antioxidant defences and damage the structure and function of cells [50]. In practical terms, this has the greatest implications for preterm neonates, particularly for their developing eyes (retinopathy of prematurity) and lungs (pulmonary dysplasia leading to chronic lung disease) [50]. For this reason, neonatal guidelines recommend targeting oxygen saturations in preterm neonates receiving oxygen [32].

However, giving more oxygen to preterm neonates is unlikely to result in higher mortality. Five large studies have compared the use of higher (91%–95%) to lower (85%–89%) oxygen saturation targeting for extremely preterm neonates, demonstrating increased mortality in the lower-saturation group (risk ratio 1.17; 95% CI 1.04–1.31) [51]. In our study, harm from excessive oxygen is even less likely because oxygen usage for neonates at the larger hospitals, where the vast majority of neonates were admitted, was already quite high at baseline, with negligible change across study periods.

Second, it is possible that implementation of our full oxygen system resulted in broader unintended consequences that may have contributed to excess mortality (e.g., distracting staff from other duties). User feedback suggests that, although healthcare workers initially found pulse oximetry to be burdensome, they felt that the full oxygen system made their work easier and reduced stress [28].

Third, it is possible that the mortality outcome was confounded by externally driven factors that we could not account for in analysis (and that were perhaps stronger for neonates than older children). In the context of the recession, these confounders may have included changed admission criteria (e.g., admitting more very sick neonates that would have otherwise been palliated or, conversely, fewer mildly unwell neonates that would have survived), care-seeking behaviours (e.g., delayed presentation, poorer birth care), care practices (e.g., not giving

antibiotics promptly, increased cost of care), or general quality of care (e.g., staff levels, motivation). If there was general confounding, then we might expect to see similar effect across all neonates (or potentially less negative effect among particular neonates who actually benefited from oxygen).

Fourth, analysis of outcomes on a hospital-by-hospital basis shows significant heterogeneity (S4 Table), and it is possible that the multifaceted intervention worked differently in different contexts. For example, H1 was a large mission hospital that maintained services relatively unchanged throughout the economic recession and realised mortality reductions for both children and neonates. In contrast, H7 was a large government hospital that provided free care to patients and recorded few deaths preintervention but was strongly impacted by the economic recession (including introducing patient fees and cutting services). H7 realised mortality increases for both children and neonates. Small study numbers in individual hospitals make definitive subgroup analysis challenging. However, preliminary exploration of this heterogeneity has identified a number of possible contextual and mechanistic factors that will be evaluated more fully in a separate paper.

We are not aware of any previous oxygen improvement studies that included neonates, and our findings raise some important questions. However, we urge caution in interpreting these data, as confounding and chance may account for some of the mortality differences we observed.

## Limitations

Using a stepped-wedge design enabled us to gain the benefits of randomisation and learn valuable lessons about implementation along the way, but it did make adjusting for temporal effects harder than if it were a parallel cluster trial. Stepped-wedge designs work best when potential confounders remain relatively stable during the study period, and this was potentially compromised by the major economic and social changes in Nigeria that likely introduced additional confounding.

We used a 2-week 'wash-in' period; however, actual practice change took longer, and oxygen use changed less than we had anticipated. A heat map helped visualise this incremental change for practice outcomes, but low death rates make heat-map figures less useful for clinical outcomes.

We included over 24,000 participants in primary analysis; however, the small number of clusters and the unbalanced cluster size limited the power of our study. This was particularly challenging for neonatal data, with the four largest hospitals contributing 84% of the data. The power to detect significant change in mortality was lower than expected because of (1) lower-than-expected baseline CFRs, (2) lower overall admission numbers, and (3) higher-than-expected ICCs. We reported the primary clinical outcome for multiple subgroups and evaluated multiple secondary practice outcomes. We used prespecified analysis methods but recognise the risk of chance findings. Both positive and negative findings should be interpreted with caution.

We used data from clinical documents using an approach similar to others conducting implementation research through clinical research networks [52]. Our use of dedicated research nurses minimised the amount of missing data, and our audit of documentation practices prior to starting the study reassured us that documentation practices overall were excellent [13]. We used standardised case definitions but did not impose additional restrictions on healthcare workers regarding their documentation. This could result in diagnostic misclassification or missed signs of severity but is unlikely to have altered our findings substantively.

## Practical implications

Our study was part of a pragmatic field trial to better inform efforts to improve oxygen access to children and newborns in Nigeria and other LMICs. We have reported results from our needs assessment and hypoxaemia prevalence estimates previously, showing that hypoxaemia is common in both respiratory and nonrespiratory childhood illnesses and that many hospitals have limited access to pulse oximetry and oxygen therapy [6, 29].

In recognition of this challenge, the Nigerian federal government recently released a national oxygen policy and strategic road map for the scale-up of oxygen services in Nigerian health facilities [53–55]. We hope that emerging findings from our project will help others seeking to improve oxygen access, and we offer the following humble suggestions.

1. Pulse oximetry is key to improving oxygen use and likely improves other aspects of inpatient care (e.g., assessment and monitoring, confidence and motivation among healthcare workers, communication and trust with caregivers). As a relatively affordable and potentially life-saving practice, pulse oximetry should be central to all efforts to improve oxygen systems and paediatric hospital care—including as a core vital sign in assessing all acutely unwell children and newborns.

2. Oxygen therapy is part of a system of care, and outcomes are influenced by many other factors. The challenges to oxygen access include many factors that also compromise other aspects of care, including weak equipment maintenance systems, poor power supplies, staff shortages, lack of clinical guidelines, and challenges of interdisciplinary cooperation. These challenges provide opportunities to use oxygen access as a means to reveal systemic weaknesses and incrementally improve the broader hospital system for improved patient outcomes.

3. Oxygen, administered as low-flow oxygen or with continuous positive airway pressure (CPAP), is an important part of neonatal care. We advise readers against making changes to neonatal oxygen policies or practices on the basis of this study and to adhere to WHO guidelines [32]. Oxygen, like any medication, should be given judiciously, accompanied by treatment of the specific medical condition and good supportive care. Future research and quality-improvement activities should seek to better understand how oxygen can be used most safely and effectively in neonatal care—particularly for the most at-risk preterm neonates.

## Conclusion

For children, there is consistent evidence across primary and secondary analyses that the introduction of pulse oximetry improved oxygen practices in Nigerian hospitals and may have reduced ALRI mortality, with no additional clinical benefit from the full oxygen system. This finding is plausible and consistent with previous studies and may indicate that pulse oximetry improves the use of existing oxygen supplies and encourages hospitals to mobilise additional oxygen supplies for paediatric use (and may improve other aspects of care).

For neonates, the interpretation is more challenging. On one hand, there was suggestion of harm for neonates when comparing the full oxygen system period to the pulse oximetry period, without a clear explanation. On the other hand, we found no effect of the improved oxygen system on neonatal outcomes compared to baseline and negligible change to oxygen use for neonates.

## Supporting information

**S1 CONSORT Checklist.**
(DOCX)

**S1 Fig. Participant flow diagram.**
(TIF)

**S1 Text. Description of participating hospitals.**
(DOCX)

**S2 Text. Analysis plan and severity-adjusted analysis model.**
(DOCX)

**S1 Table. Description of the improved oxygen system intervention.**
(DOCX)

**S2 Table. Timetable of project implementation.**
(DOCX)

**S3 Table. Primary outcomes subgroup analyses.**
(DOCX)

**S4 Table. Primary clinical outcomes by individual hospital.**
(DOCX)

## Acknowledgments

We thank the participating hospitals and their clinical, technical, and managerial staff: Adeoyo Maternity Hospital (Ibadan, Oyo state); Baptist Medical Centre (Saki, Oyo state); Mother and Child Hospital (Akure, Ondo state); Oluyoro Catholic Hospital (Ibadan, Oyo state); Oni Memorial Children's Hospital (Ibadan, Oyo state); Our Lady of Fatima Catholic Hospital (Osogbo, Osun state); Sacred Heart Hospital (Abeokuta, Ogun state); Seventh Day Adventist Hospital (Ife, Osun state); State Hospital Ijaye (Abeokuta, Ogun state); State Hospital Oyo (Oyo, Oyo state); State Hospital Saki (Saki, Oyo state); State Specialist Hospital (Akure, Ondo state). We thank all members of the Nigerian Oxygen Implementation team; representatives from the Federal Ministry of Health; representatives from the State Ministry of Health and hospital management boards in Oyo, Ondo, Ogun, and Osun states; and support staff at the Centre for International Child Health in Melbourne, Australia.

## Author Contributions

**Conceptualization:** Hamish R. Graham, Adejumoke I. Ayede, Shamim A. Qazi, Rasa Izadnegahdar, Adegoke G. Falade, Trevor Duke.

**Data curation:** Hamish R. Graham, Ayobami A. Bakare, Oladapo B. Oyewole, Eleanor F. G. Neal, Adegoke G. Falade.

**Formal analysis:** Hamish R. Graham, Eleanor F. G. Neal, Cattram D. Nguyen, John B. Carlin.

**Funding acquisition:** Hamish R. Graham, Rasa Izadnegahdar, Adegoke G. Falade, Trevor Duke.

**Investigation:** Hamish R. Graham, Ayobami A. Bakare, Adejumoke I. Ayede, Amy Z. Gray, Barbara McPake, David Peel, Oladapo B. Oyewole, Shamim A. Qazi, Adegoke G. Falade, Trevor Duke.

**Methodology:** Hamish R. Graham, Ayobami A. Bakare, Adejumoke I. Ayede, Amy Z. Gray, Barbara McPake, David Peel, Olatayo Olatinwo, Cattram D. Nguyen, Shamim A. Qazi, John B. Carlin, Adegoke G. Falade, Trevor Duke.

**Project administration:** Hamish R. Graham, Ayobami A. Bakare, Adejumoke I. Ayede, Olatayo Olatinwo, Eleanor F. G. Neal, Adegoke G. Falade, Trevor Duke.

**Resources:** Adegoke G. Falade.

**Supervision:** Hamish R. Graham, Ayobami A. Bakare, Adejumoke I. Ayede, Amy Z. Gray, Barbara McPake, David Peel, Olatayo Olatinwo, Oladapo B. Oyewole, Adegoke G. Falade, Trevor Duke.

**Validation:** Hamish R. Graham, Ayobami A. Bakare, Cattram D. Nguyen, Trevor Duke.

**Visualization:** Hamish R. Graham.

**Writing – original draft:** Hamish R. Graham.

**Writing – review & editing:** Hamish R. Graham, Ayobami A. Bakare, Adejumoke I. Ayede, Amy Z. Gray, Barbara McPake, David Peel, Olatayo Olatinwo, Oladapo B. Oyewole, Eleanor F. G. Neal, Cattram D. Nguyen, Shamim A. Qazi, Rasa Izadnegahdar, John B. Carlin, Adegoke G. Falade, Trevor Duke.

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
