## [Decision Letter · Decision Letter 0]

18 Jul 2019

Dear Dr. Graham,

Thank you very much for submitting your manuscript "Oxygen systems to improve clinical care and outcomes for children and neonates: a stepped-wedge cluster-randomised trial." (PMEDICINE-D-19-02095) for consideration at PLOS Medicine. 

[LINK]

In light of these reviews, I am afraid that we will not be able to accept the manuscript for publication in the journal in its current form, but we would like to consider a revised version that addresses the reviewers' and editors' comments. Obviously we cannot make any decision about publication until we have seen the revised manuscript and your response, and we plan to seek re-review by one or more of the reviewers. 

We expect to receive your revised manuscript by Aug 08 2019 11:59PM. Please email us (plosmedicine@plos.org) if you have any questions or concerns.

We look forward to receiving your revised manuscript. 

Sincerely,

Clare Stone, PhD

Managing Editor 

PLOS Medicine

plosmedicine.org

In the abstract and elsewhere, you report "no effect ... on death" for children with the oxygen system (CIs cross unity), and "may have increased ... neonatal odds of death" (CIs do not cross unity). So, it may appear as though you believe a negative result but are more sceptical about a positive result (of an undesirable outcome). Please address. 

I'd suggest adopting less equivocal language ... i.e. noting an observed increase in risk of death in the neonatal group, which you can then discuss possible reasons for. 

Please be more explicit about the limitations of the study in the abstract – starting the final sentence of the Methods and Findings section with ‘ Limitations of our study are…’

Please provide p val;ues with Cis in the abstract

Data – can you please clarify is all users can access data at osf.io/pumdq and can you please provide a direct link to the data set in this location if it’s a large depository?

Please use square brackets for refs in the main text

Did your study have a prospective protocol or analysis plan? Please state this (either way) early in the Methods section.

c) In either case, changes in the analysis—including those made in response to peer review comments—should be identified as such in the Methods section of the paper, with rationale.

Please provide a flow chart of the trial participants, arms and exclusions etc, per standard for trial manuscripts. Apologies if I am missing this. 

Please provide a CONSORT reporting checklist – as a supp file. 

Comments from the reviewers:

Reviewer #1: The authors are to be congratulated on addressing a very important bur relatively neglected topic, namely how best to use oxygen in the care of sick children in low or middle income countries. The results of the trial are rather surprising, especially the possible increase in neonatal mortality in the oxygen intervention group, which may have been due to chance, but could be true in which case this is an important and rather alarming finding which needs further investigation.

Major points 

The trial has a number of weakness which are largely covered in the discussion, such as the fact that it was under-powered to detect a small difference between groups, but I think that there are a couple of additional ones that could be mentioned in the discussion. 

a. The trial design. A stepped wedge design was chosen rather than a contemporary, randomised study for sound logistic reasons but step wedged trials only work well when risk and potential confounders remain relatively stable throughout the period of the study. For reasons outside the control of the investigators, this was not the case in this study, and this could have influenced the outcome. An attempt to take this into consideration has been made through use of a severity score but this is still a concern.

b. Heterogeneity between hospitals. Facilities and numbers of trained staff varied widely between hospitals, for example between hospitals H1 and H9. An attempt was made to overcome this problem in matching the intervention groups through stratification but this does not take account of the fact that the oxygen intervention might have had variable impacts in hospitals with well or poorly developed oxygen supplies, infrastructure and staff. An attempt to look at this issue is given by the presentation of results from each hospital but numbers were too small to undertake a definitive sub-group analysis. This issue could be mentioned in the discussion.

Other points

a. Pre-intervention data. Pre-intervention data were collected retrospectively whilst in the intervention periods they were collected prospectively using a standard CRF. Data from the pre-intervention period are likely to have been much less complete, and perhaps less reliable, than that that collected prospectively. How was this issue handled?

b. Impact on clinical performance. Introduction of the oxygen system was accompanied by training (appendix 2)) but it is not clear how focused this was and whether there was some additional, more general training provided on management of the sick child which could have influenced outcome in addition to the input of oxygen therapy.

c. Role of initial hypoxaemia. Was a sub-analysis done of impact on children who were hypoxaemic at the initial assessment or were the guidelines on when to give oxygen followed so assiduously that it was not possible to do this?

d. Sustainability. Do the investigators have any information on what has happened in the trial hospitals since the study finished? Has there been any long-term impact or has the situation reverted to pre-intervention days? 

e. Messages. A message can be disseminated to paediatricians working in LMIC that results from this study support the findings from a limited number of previous studies (Laos, PNG) that giving oxygen to children with severe pneumonia reduces mortality. However, it is not clear what the message about neonates should be. Should they not be given oxygen? How could this issue be clarified? Some guidance from investigators who have made a major investment in this area of paediatric research and care would be helpful. 

Reviewer #2: Statistical review

This paper reports a stepped wedge cluster randomised trial evaluating the rolling out of an intervention for improving hospital oxygen systems for treatment children with acute lower respiratory infection. The rolling out was in two phases: pulse oximetry and then multi-faceted oxygen system. The primary analysis is full vs pulse, although both are compared to baseline also.

The trial had a number of practical issues occur beyond the control of the authors. This may have caused more than usual confounding. I think the authors do an excellent job of discussing this and the potential implications in light of the unexpected results from the trial. Generally the statistical methods and reporting is also very good quality. I have some comments, listed below:

1. Line 51 (and line 289) 'no effect on odds' - 'no significant association' or 'no evidence of effect on odds'. Since this was a trial testing a specific hypothesis, I recommend p-values are provided in addition to CIs.

2. Lines 54-56: provide effect size and CI.

3. Abstract and results: personally I think p-values can add useful information when reported in addition to CIs: this is clearly a prospectively planned study testing pre-specified hypotheses rather than an exploratory one where p-values might be inappropriate. However the published protocol mentions just effect sizes and CIs would be reported so the lack of p-values is consistent with what was planned.

4. Line 57: "Our study had lower than anticipated power to detect change in mortality outcomes." - briefly say why? Lower event rate than anticipated? Unclear whether the next sentence is the explanation but if a link between this and the reason for lower power could be given, this would be clearer.

5. Intervention section - can more be said in this section about the pulse oximetry component and how much earlier that was available compared to the full intervention?

6. I was a bit confused by the use of 'mortality' as the primary outcome and then 'pneumonia CFR' in the power calculation. Are these are completely the same? I'd have thought that all-cause mortality and pneumonia CFR might not be completely identical. 

7. Sample size calculation: can the assumed rate of pneumonia CFR in the pulse oximetry and full intervention period be provided?

8. Line 226-227: can a bit more detail about the random hospital-time interaction effects be added? It was not clear how this was done.

9. Line 291-292: although I agree the broad conclusions are the same the severity adjusted model seems to have substantially strong evidence of an increase in odds for neonates.

10. Is table 3 needed when it seems like all the information is included in table 4?

James Wason

Reviewer #3: General comments

This is a peer review of the manuscript entitled "Oxygen systems to improve clinical care and outcomes for children and neonates: a stepped-wedged cluster randomised trial" by Graham and colleagues. This report is of results from a stepped-wedge cluster randomized trial evaluating the effects of oxygen system introduction in 12 Nigerian hospitals on mortality and clinical practices.

This is an extremely important topic and the question addressed is also extremely important. I want to first off acknowledge the major efforts that were undertaken to conduct this trial and the incredible challenges that were faced to execute this work. Hats off. For me there are 5 main observations after reading and reflecting on this important work.

1) Due to an unfortunate miscalculation the study was substantially underpowered for its main outcomes of oxygen systems implementation on hospital mortality (35-50% power for the effect sizes listed). Thus the main findings of no difference in mortality among children between the pulse oximetry and oxygen system time periods is not surprising. It is notable though that the authors found a surprising increased odds of mortality among neonates, but they have also acknowledged that given the lack of power this finding also may be due to chance and should be interpreted with caution. This neonatal outcome is potentially very important and deserves attention.

2) Due to unfortunate instability in Nigeria there were major health system disruptions that could not really be accounted for in the analyses. It appears that roughly half of the hospitals were in the pulse oximetry period during this instability (and this period has 4,000+ less observations than the oxygen system period), which may have further compromised this period and falsely lowered mortality during this period (thus confounding results). This raises further uncertainty to any findings and additionally complicates having limited power to start with (likely further lowered power also).

3) Secondary exploratory outcomes showed lower odds of death for ALRI only among children in the pulse ox and oxygen system periods, compared to the pre-intervention period; but misclassification could be an issue in the pre-intervention period since this data was collected differently (retrospective) (the authors state in the limitations section that misclassification shouldn't be an issue based on audits)? I would encourage the authors to include the audit data in their report to support this as it would strengthen their findings that the pre-intervention period is valid and comparable to the intervention period. 

4) Some key results, and the ensuing interpretation, seem to counter one another, or are at least not completely supportive (or maybe I'm not understanding them fully)? Specifically, secondary outcome analyses showed that compared to the pre-intervention period, the pulse oximetry period and the full oxygen period both reduced the odds of death for children with ALRI. But the practice outcomes analysis shows that while pulse oximetry practices improved dramatically, the pulse oximetry period DID NOT have any impact on overall oxygen use or perhaps even more importantly DID NOT have any impact on oxygen use for those with hypoxemia. I'm having a hard time fitting this together into a coherent story given the children most likely to benefit would presumably be those that are hypoxemic.

On the other hand, the practice outcomes also show that oxygen practices improved (but to a lesser extent than pulse oximetry practices) when the full oxygen system was introduced, and this could make sense. But there must be something else happening to explain why pulse ox alone reduced mortality when it didn't change oxygen use, especially among those most in need of it? I could not find any conclusive explanation in the data presented, but I fully understand the challenges faced during this study.

5) I would like the authors to include much more analyses on the outcomes of children and neonates with hypoxemia. I find this critical subgroup missing throughout and given the population most likely to benefit from the intervention are hypoxemic children and neonates this should be presented with the main results. (in my view these children and neonates should have been the target population for the primary outcome). This data should be available given the pulse oximetry period involved pulse oximetry introduction and this data should be included with this report. Please also include data on the quality of pulse oximetry measurements. Please also clarify why older children aged 5-14 or even aged 2-5 years were included in the primary analysis given most mortality is experienced among children less than 2 years of age? Inclusion of this older population with less mortality may have further diluted the mortality effect of the oxygen system possibly but this is not discussed anywhere.

Abstract:

No substantive comments.

Main report:

Introduction:

Well written. No comments.

Methods

Page 5, participants section, lines 129-130. Please clarify what is meant by AGF and AIA enrolled hospitals. Not clear.

Page 5, participants section, lines 133-135. How was the 2 week wash in period determined? A priori? Based on certain indicators? Please clarify.

Page 6, Intervention section, lines 153-154. How are hospitals consented? Please clarify.

Page 6, Procedures section. How much support was provided during the pulse oximetry period? How much training, supervision etc.? Was this comparable to the full oxygen system support? I'm assuming that training regarding the use of pulse oximeters and target population for measurement were included in the distribution of the devices?

Page 7, outcomes, lines 177 to 180. Given oxygen would be most beneficial (presumably) to children who are hypoxemic, why did the authors not choose hypoxemic children and neonates as their source population given I would assume this is the main target population implementation of oxygen systems would aim to benefit? The main analysis was to compare pulse ox to the full oxygen system such that the oxygen saturation levels should be recorded and known for children during these two primary time periods. Please see more detailed comment above.

Results

Page 9, lines 260-262. Could the presentation of more severe illness during the pulse oximetry and full oxygen periods be attributed to data quality differences between the prospective and retrospective periods, especially since support was provided during the prospective data collection period to some degree and this may have influenced the practice of recording of clinical signs (hospitals were not masked)? Please include any audit data supporting the notion that the pre-intervention period data is valid.

Page 10, fidelity and variation from the protocol section, lines 268-269. Given there were power supply issues during the full oxygen implementation period, was any data collected recording periods of power interruptions (frequency and length) and outcomes during these interruptions. Could such interruptions be accounted for in the analyses? 

Page 11, same section, lines 278-279. Please clarify what the costs were for oxygen, whether this varied by hospital, and whether this could/should be accounted for in the analyses? (or maybe it was through the random effects)

Page 11, lines 281-283. How did the analysis account for patients who left against medical advice? Presumably this may mean that hypoxemic children at high risk of death left the hospital early and died at home, and these outcomes could have been missed?

Intervention effects, primary outcomes, page 10. Given the study was so under-powered it is not surprising that the study failed to reject the null hypothesis and its findings were inconclusive. Please add the severity adjusted analyses OR and aOR to the results text here even though there was no qualitative difference to the basic model.

Given the initial study design intended to compare pulse oximetry to the full oxygen system, why is the primary outcome of death among all children, children with ALRI, and neonates and preterm regardless of their oxygen saturation? Shouldn't the primary target population be children who are hypoxemic? Please see more detailed comment above.

Page 11, practice outcomes, lines 304-305. Please explain or expand on this sentence indicating that pulse oximetry practices improved by oxygen practices did not.

Page 11, practice outcomes, lines 306-307. Same comment as above. Please clarify this sentence. How specifically did practices improve or not improve. 

Discussion

Lines 347 to 352, page 12. My main concern with drawing these conclusions is the likelihood of misclassification and also that the practice outcomes seem to be counter to these conclusions. Given the study relied on medical chart extraction and assumed that if a clinical sign was not documented that it was not present, there is likely to be variability in documentation of these signs across the study time period. I would assume that the pre-study period such documentation would be even more variable. I think the authors need to somehow address the potential for misclassification as an important bias in their results, especially the analyses that are using pre-study data that was retrospectively collected. It is further difficult to understand how pulse oximetry would improve outcomes alone when the authors are also reporting data that oxygen use for hypoxemic children did not improve during the pulse oximetry period for both neonates and children. Having a hard time putting all of these pieces together. 

Did the authors do any validation exercises of the clinical medical chart to see how reliable the data actually was? This would have been helpful to understand data quality and likelihood of misclassification.

Lines 357 to 360 (and the text following this) in the discussion seem to be directly counter to lines 307 to 309 and lines 311 to 313 in the results section. Please clarify the differences.

Lines 383+ "clinical impact for neonates" section. Very interesting findings. Important.

[LINK]

---

## [Decision Letter · Decision Letter 1]

10 Sep 2019

Dear Dr. Graham,

Thank you very much for re-submitting your manuscript "Oxygen systems to improve clinical care and outcomes for children and neonates: a stepped-wedge cluster-randomised trial." (PMEDICINE-D-19-02095R1) for review by PLOS Medicine.

I have discussed the paper with my colleagues and the academic editor and it was also seen again by reviewers. I am pleased to say that provided the remaining editorial and production issues are dealt with we are planning to accept the paper for publication in the journal.

[LINK]

We look forward to receiving the revised manuscript by Sep 17 2019 11:59PM. 

Sincerely,

Clare Stone, PhD

Managing Editor 

PLOS Medicine

plosmedicine.org

Requests from Editors:

Please add country to title.

Abstract – “This paper reports the effects of an Background: Improving oxygen systems may improve clinical outcomes for hospitalised children with acute lower respiratory infection (ALRI). This paper reports the effects of an improved oxygen system on mortality and clinical practices in 12 Nigerian hospitals..” (I think some odd cut and pasting has occurred, please amend)

Abstract – 12 hospitals – all in same city/ region and are they maternal / neonatal or general hospitals?

Data statement in submission form, please change to All dataset files are available from the OSF database: osf.io/pumdq

Analysis plan?

Line 161 – participants – again, pleae provide more information about the location of hospitals and where patients are recruited from in terms of regions / cities.

Written consent from parents/ careers? 

Please switch Tables 1 and 2 so participant data is listed first and alter any call outs in the text. 

CONSORT checklist – thank you for providing; however please use sections and paragraph numbers instead of pages as these can change after production changes and formatting. 

- at line 53, "associated with an increased risk of death" might be better than "positively associated with death". 

- at line 86, it might be better to write "there was an increase in neonatal deaths" rather than quote the 45% figure. 

- the Lancet-style "role of the funding source" can be removed from the methods section – this is automatically pulled in from the funding statement in the submission form.

- the wording at lines 340-2 does not seem right (I'd suggest "... was associated with an increased risk of death for neonates (numbers) but not for preterm/LBW neonates (numbers) ..."). 

- similarly, I'd suggest changing the wording at lines 350-351 (e.g. "... system interventions were not associated with risk of death for children with ALRI ..."); "in contrast" would need to be removed at line 353. 

- at line 361, I'd suggest adapting the text to "... there was a non-significant increase in overall oxygen use ...", followed by "and there was no evidence ...". 

- the start of the discussion needs to be restructured so as to begin with a summary of the findings

- at line 454, I'd suggest changing "more weakly indicated" to "... although there was no significant increase among preterm/LBW neonates ...".

- update ref 6, or please provide acceptance letter

- ref 47 can be trimmed

Comments from Reviewers:

Reviewer #1: This referee thanks the authors for their detailed responses to his comments on the earlier version of the paper which have all been addressed as far as possible. This paper rep[orts an important study.

Reviewer #3: All of my concerns/comments have been fully and thoroughly addressed by the authors. Sincere thank you and congratulations.

[LINK]

---

## [Editor Report · Decision Letter 2]

11 Oct 2019

Dear Dr. Graham, 

On behalf of my colleagues and the academic editor, Dr. Brian Greenwood, I am delighted to inform you that your manuscript entitled "Oxygen systems to improve clinical care and outcomes for children and neonates: a stepped-wedge cluster-randomised trial in Nigeria." (PMEDICINE-D-19-02095R2) has been accepted for publication in PLOS Medicine. 

PRODUCTION PROCESS

PRESS

PROFILE INFORMATION

Thank you again for submitting the manuscript to PLOS Medicine. We look forward to publishing it. 

Best wishes, 

Clare Stone, PhD

Managing Editor 

PLOS Medicine

plosmedicine.org